# A Continuous Motion Shape-from-Focus Method for Geometry Measurement during 3D Printing

**DOI:** 10.3390/s22249805

**Published:** 2022-12-14

**Authors:** Jona Gladines, Seppe Sels, Michael Hillen, Steve Vanlanduit

**Affiliations:** 1Faculty of Applied Engineering, University of Antwerp, 2020 Antwerp, Belgium; 2Department of Mechanical Engineering, Pleinlaan 2, Vrije Universiteit Brussel, 1050 Brussel, Belgium

**Keywords:** focus variation, shape measurement, shape-from-focus, laser triangulation, topography, optical dimensional metrology

## Abstract

In 3D printing, as in other manufacturing processes, there is a push for zero-defect manufacturing, mainly to avoid waste. To evaluate the quality of the printed parts during the printing process, an accurate 3D measurement method is required. By scanning the part during the buildup, potential nonconformities to tolerances can be detected early on and the printing process could be adjusted to avoid scrapping the part. Out of many, shape-from-focus, is an accurate method for recovering 3D shapes from objects. However, the state-of-the-art implementation of the method requires the object to be stationary during a measurement. This does not reconcile with the nature of 3D printing, where continuous motion is required for the manufacturing process. This research presents a novel methodology that allows shape-from-focus to be used in a continuous scanning motion, thus making it possible to apply it to the 3D manufacturing process. By controlling the camera trigger and a tunable lens with synchronous signals, a stack of images can be created while the camera or the object is in motion. These images can be re-aligned and then used to create a 3D depth image. The impact on the quality of the 3D measurement was tested by analytically comparing the quality of a scan using the traditional stationary method and of the proposed method to a known reference. The results demonstrate a 1.22% degradation in the measurement error.

## 1. Introduction

Additive manufacturing is a constantly developing production method, with a lot of effort going into zero-defect manufacturing [1]. Nowadays, most 3D additive manufacturing machines, also known as 3D printers, are mainly open-loop systems [2,3,4]. This can result in the scrapping of parts due to defects that only become apparent when the printing process is complete and the part is thoroughly inspected. Early detection of defects or abnormalities could save material and parts. For accurate inspection and defect detection, a precise 3d scanning method is required. Recently, some studies have closed the printing loop by employing state-of-the-art 3D reproduction methods, such as laser triangulation [5,6,7] or fringe projection [8,9,10]. Shape-from-focus (SFF) [11,12], also known as focus variation microscopy (FV), is a method that is also capable of measuring at μm level accuracy and precision [13,14]. It has an advantage over laser triangulation and fringe projection, in that it can also deliver an all-in-focus texture map of the surface, making it easy to detect surface defects. It has already been presented as an on-machine solution for accurate surface topography measurements by Santoso T. et al. [15]. Shape-from-focus is based on the relationship between object–camera distance and the focus setting of the camera-lens. An object point at a certain distance from the camera will only be in focus at a specific focus setting of the lens, especially when imaging with a narrow depth of field (DOF) (Figure 1).

By capturing multiple images at different focus distances using a tunable lens and estimating the amount of pixel focus through a focus measure operation [12,16], the object–camera distance for every pixel can be accurately determined, effectively creating a precision depth-map. Shape-from-focus has already successfully been applied to additive manufactured parts in offline quality inspections [17]. The traditional shape-from-focus method is still being researched [18,19]. However, state-of-the-art implementations cannot be applied for inline measurements. An SFF measurement requires tens of images to be captured with the 3D-printed part at the same position relative to the camera/lens to produce a depth-map. Precision measurements of larger surfaces using SFF thus require the stitching of multiple discrete measurements, while the nature of 3D printing is to print in one continuous motion.

In this work, a method is proposed that adapts the current SFF technique into a scanning method to allow for larger areas to be measured to the same accuracy and precision while printing (Figure 2). By moving the camera or sample during a repeated scan across the focus range of the SFF setup, essentially the same data as with a discrete measurement can be collected. However, the data is organized differently: images will be shifted and the information can be split across multiple images. Through the use of image stitching and aligning methods, the gathered data can be re-arranged for processing with current SFF methods. The following section describes the setup that was used and how to control it, the methods that were employed for processing and how the results are validated.

## 2. Design of the Setup and Experiment

With current direct energy deposition (DED) additive manufacturing machines printing at resolutions below 50 μm, to do in-line inspection, one would need a system capable of measuring with a precision of 25 μm or below. The typical scanning speeds for DED systems are in the range of 5 mm/s to 20 mm/s [20]. Thus for in-line inspection, one would need a system capable of measuring in that speed range. The following sections describe the measurement setup that was built for this purpose and the development of the proposed scanning SFF method.

### 2.1. Measurement System Setup

The measurement setup is presented in Figure 3. It allows for precise depth profile measurements using shape-from-focus. The setup consists of three main parts: the camera (1.3MP Mako U-130B Mono) with electronically tunable lens (Optotune EL-10-30C [21]) and finite corrected objective lens (Mitutoyo 3x Objective CF), a translation stage (Zaber X-LSQ300B) and a signal generator (NI USB-6343DAQ). The camera with tunable lens and objective lens provides a 9 mm by 11 mm field of view (FOV) with a narrow depth of field for use with shape-from-focus. The focus can be controlled over a range of 10 mm. The combination of optics and camera results in an optical resolution of approximately 25 μm. The translation stage allows for larger areas to be measured and the signal generator provides a trigger signal to the camera as well as an analog signal to control the tunable lens. The lighting consists of a ring light and additional LED light to provide some oblique illumination. Adding oblique illumination provides extra contrast on the 3D-printed surface for the FMO to work. The tunable lens allows for varying the focus distance in order to create a stack of 1024 by 1280 pixel images at different and equally spaced focus distances.

As explained in Section 1, in the traditional sense of the shape-from-focus, when imaging a larger field of view, the camera is placed at different discrete positions and the individual measurements are stitched together. In the proposed method, the camera or sample is constantly moving. Measuring with the shape-from-focus at speeds of, e.g., 15 mm/s and above requires a fast control of the focus distance. Traditional shape-from-focus uses a precision translation stage to change the focus distance. However, these precision translation stages usually have a maximum speed of about 2–3 mm/s [22]. The proposed method would require speeds over 30 mm/s. Therefore, an electronically focus tunable lens can be used. Such a lens can be controlled with frequencies of 50 Hz and above, which, with a focus range of 10mm, corresponds to 500 mm/s in our setup. To use the lens at these speeds, it must be controlled with an analog signal. Since the exact focus distance must be known for every captured image, the camera must be hardware triggered by a trigger signal that is synchronized with the lens control signal. Both signals are supplied by the DAC.

The controller of the tunable lens accepts an analog signal between 0 and 5 V to control the current to the lens over a range of 292 mA. This analog signal is sampled at 10 bit, thus allowing for a maximum of 1024 different steps in the current control range of the lens. Ideally, one would control the lens with a sawtooth signal and trigger the camera with a square wave signal. However, this would require perfect synchronisation between the lens signal and the actual trigger of the camera. Since the delay between the trigger signal applied to the camera and the actual capturing of the image is unknown, one cannot trust it to be constant. Therefore, we chose to control the lens with a staircase signal with the number of steps being equal to the number of images required in a focus stack. The camera is then triggered by the falling edges of a square-wave signal. The falling edges are centred onto the steps of the staircase signal to the lens driver. Both signals are presented in Figure 4.

The frequencies of the staircase signal and the trigger signal depend on the translation speed of the camera or sample. The maximum translation speed in mm/s is defined by a few parameters: The FOV of the measurement system, the frame rate of the camera, the amount of frames in a stack and the amount of overlap between consecutive stacks. The proposed method requires that a portion of the frames overlap to allow consecutive depth maps to be stitched together. In the time it takes the camera to create the amount of images for one depth map calculation, the camera cannot translate more than 50% of the width or height of the FOV, depending on the movement direction. The theoretical maximum amount of the translation $l_{max}$ is given by Equation (Equation 1), where *w* is the width or height of the FOV.
(1)$$l_{max}=\frac{w}{2}$$

For example, an imaging system with an FOV of 9 mm by 11 mm, the theoretical maximum amount of translation between focus stacks is 5.5 mm if translating in the direction width of the FOV and 4.5 mm if translating in the direction of the height of the FOV. This maximum translation distance is not usable if measurements need to be stitched. Thus, a portion of consecutive focus stacks must overlap. The amount of translation with respect to the amount of overlap is given by:(2)$$l=\frac{w-(w\cdot P)}{2}$$
where *P* is the percentage of overlap. Knowing the maximum frame rate of the camera *r* and the number of images required in a focus stack *N*, one can calculate the maximum translation speed *s* for a measurement with:(3)$$s=(\frac{r}{N})\cdot l$$

From these equations, one may notice that a larger FOV allows for faster translation speeds while measuring at a lower optical resolution. The measurement speed can also be increased by employing a higher imaging frame rate or by reducing the number of images required in a focal stack. The number of images that is required can be reduced by using various interpolation methods [11,23,24]. An example is given for the maximum translation speed with our setup.

A camera with a FOV width of 11 mm and a frame rate of 150 fps can capture 50 frames in 0.333 s, resulting in a theoretical measurement speed of 15 mm/s, given a 9.1% overlap between focus stacks. Doubling the frame rate of the camera to 300 fps or reducing the number of required images in a stack to 25 allows the system to measure at 30 mm/s.

The next step in the measurement method is the processing of the images to a depth-map, which is described in the following section.

### 2.2. Image Processing

Traditionally, with the shape-from-focus, the stack of differently focused images are all captured with the object and camera at a fixed relative position; thus, no image alignment is required. In this paper, we consider the case where the part moves relative to the camera during the SFF measurement process. This means that the recorded images in the SFF measurement process are shifted relative to each other. This required the images to be properly registered and aligned before the focus measure operator (FMO) can be applied to build the depth map.

The alignment is based on the integer pixel shift calculated from the translation speed. For example, if one is measuring while the object is moving at 15 mm/s with a stack of 50 images captured in 0.333 s, each image is shifted by approximately 0.1 mm with respect to the previous image, which translates to 11.64 pixels. As a result, the total image shift from top to bottom is 4.995 mm or 582 pixels. To align the images, the amount of shift in pixels for each image is thus calculated back from the speed of the translation and that shift is then inversely applied to the image (Figure 5).

After the registration process is completed, the images are ready for a traditional focus measure operation and conversion to a depthmap [25]. However, since the object translated over 50% of the FOV at most, only 50% of the focus stack can deliver useful depth information. Therefore, images from the multiple focus stacks need to be stitched for a larger FOV. When translating over slightly less than 50% of the FOV, there will be some overlap between images of consecutive stacks. Phase correlation can be used for registration and stitching [26,27]. Registration on the frames with only a few pixels in focus proved difficult at full resolution. By first reducing the resolution of the images to a quarter of the original resolution, the correct transformation matrix for registration could be calculated. Figure 6 presents the process of stitching 2 stacks before applying the focus measure operation.

### 2.3. Measurement Target

The final part of the measurement setup is the measurement target. A specific target for these measurements has been designed based upon a portable characterization target (PCT) [28] (Figure 7). It measures 100 mm by 100 mm and uses different shapes and features to do a complete characterisation of a measurement system. The target was printed in TiAl6V4 to have a representative surface for 3D printing in terms of surface roughness, colour and reflectivity.

### 2.4. Experimental Setup

In order to validate whether the proposed method has an impact on the quality of the produced depth maps, an experiment is designed. One of the features of the PCT, a tetrahedron with 10 mm base and 8.165 mm height, is measured with the camera in a stationary position as well as in a scanning motion using the proposed method (Figure 7).

The quality comparison between the depth maps is then executed based upon the analytical measures Root-Mean-Square-Error (RSME), Correlation Coefficient (CORR) and Peak-Signal-to-Noise-Ratio (PSNR) [29,30,31,32]. Both depth-maps will contain the same sections of the calibration target and will be compared to a reference. A reference for the exact form of the 3D-printed tetrahedron is not available, so a true comparison is difficult. It would require a higher precision instrument to obtain an accurate reference measurement, and that measurement would also not be completely free of measurement errors. Therefore, we opted to use a depth map generated from the calibration target’s CAD file for the analysis. Comparing the measurements to the CAD design of the part does not result in the true errors introduced by the measurement method. The comparison will also include the deviations in the 3D-printed PCT caused by the printing process. Since these deviations are equal for both measurements, one can assume that the degradation of the depth map quality, between the stationary measurement method and the scanning method, can be attributed to the measurement method. The results of this experiment are discussed in the following section.

## 3. Results and Discussion

Figure 8 shows the recovered depth-maps of both the stationary method as well as the proposed scanning method. Compared to the ground truth (GT), both maps look very similar. However, on the map from the scanning SFF method, more noise is visibly noticeable and some warping on the edge of the tetrahedron can be observed. This will result in some degradation compared to the stationary SFF method.

From the graph in Figure 9, it is clear that the proposed method causes some additional degradation. The graph represents the analytical comparison between the ground truth, which is an extract from the CAD design, the stationary SFF method (1), and the proposed scanning method (2). Since the comparison is made to the CAD design, the measured degradation is, as explained, a combination of errors introduced by the 3D printing process of the PCT and errors introduced by the measurement. Since the errors from the 3D printing process and the optical aberrations of the measurement system are the same for both measurements, the degradation between the stationary SFF and scanning SFF measurement shows the degradation due to the new measuring method. The root-mean-square-error (RSME) degrades by 0.1 mm or 1.22%, relative to the height of the object, and the correlation between the measurement and the reference degrades by just 0.62%. This shows that the impact of the method is considered small.

Figure 10 contains the distribution of errors for each measurement point from the two measurements divided into six groups. From the bar graphs, one may notice that the proposed method introduces some additional errors, because there are more points with a higher deviation from the GT.

Looking at the difference map between the measurements and the reference (Figure 11), one may notice that most deviations with the proposed method are introduced at one side of the measurement. A possible cause of this could be the lighting that changes slightly during a scanning measurement, another possible cause could be the optical distortion of the microscope objective. More research is needed to discern the root cause of these deviations.

These results suggest that the proposed method can potentially be used for the in-line control of 3D printers. However, like any other optical metrology solution, it has some limitations. Due to it being an optical method, occlusions can be an issue. However, because the optical axis of the camera is parallel to the extruder, occlusions with SFF are less of a problem than, for example, with laser triangulation or structured light profilometry, where the optical axis of the camera, the projector or the laser are angled compared to the extruder. The method can thus only be used during printing, as overhangs are not measureable once printing is finished. Another limitation of the method is the limited height measurement range. For μm precision measurements, the FOV needs to be small. Due to the optical design of most microscope objectives, higher magnification leads to a reduced focusing range of the ETL. Thus, the measurement resolution and range will always be a trade-off. The last major limitation is the measurement speed. Although capable of measuring at speeds of 15 mm/s with the proposed components, for some applications this is not fast enough. As explained in Section 2.1, measurement speed and precision are also a trade-off.

## 4. Conclusions

A new method was proposed and introduced for using the shape-from-focus in continuous motion, such that it may be used for in-line additive manufacturing inspection. It uses a fast tunable lens controlled by an analog signal, combined with a hardware-triggered camera. The method was tested by comparing the quality of a measurement using the proposed method with a stationary measurement, both measured with the same system. The measurement results demonstrate minor degradation (1.22%) in measurement quality by applying the proposed method. From this, we can conclude that the method could be applicable for in-line inspection in 3D printing applications. Yet, further research is needed to validate the influence of additive manufacturing process parameters such as CAD design, part orientation and print material on measurement accuracy and precision. The proposed method could also potentially be used for other applications, such as inspections of movable objects on conveyor belts or by extension any situation where the object is moving relative to the camera. Although this work proves that the method works, it has some practical limitations. First and foremost, the trade-off between speed and measurement precision. Secondly, occlusions may hinder part measurement, as is the case with any optical inspection method. Lastly, the depth measurement range is mostly small compared to other 3D profilometry methods. Another item to consider for future research is the possibility to improve the quality of the measurement by improving the alignment of the images in the focus stack. The current alignment is coarse because the motion of the translation stage that was used for the measurement is non-constant. This non-constant speed is introduced by the use of a stepper motor. The alignment can potentially be improved by using phase correlation or other registration techniques.

## Figures and Tables

**Figure 1 sensors-22-09805-f001:**
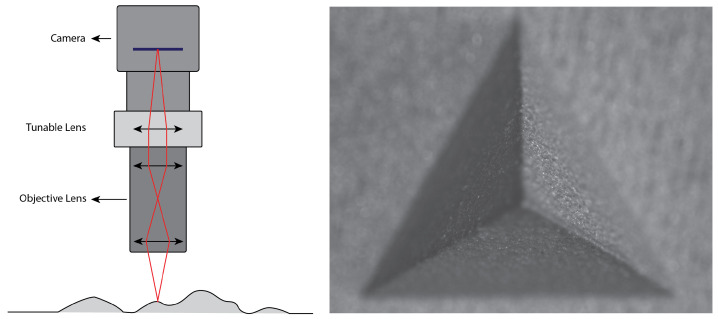
**Left**: a schematic representation of a tunable lens-based shape-from-focus system. **Right**: an image with narrow depth of field from the shape-from-focus system that was used in this research.

**Figure 2 sensors-22-09805-f002:**
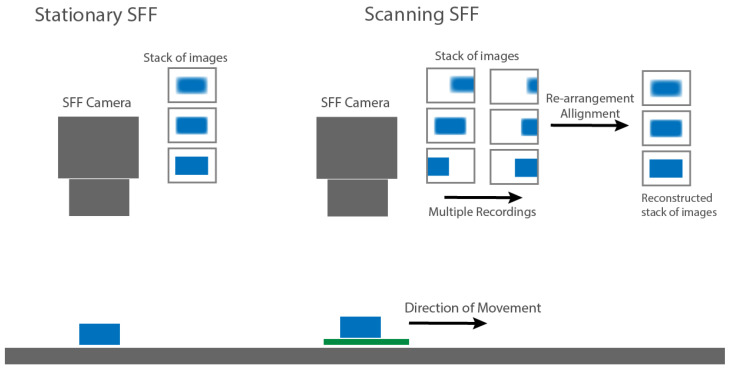
The concept of scanning SFF compared to stationary SFF.

**Figure 3 sensors-22-09805-f003:**
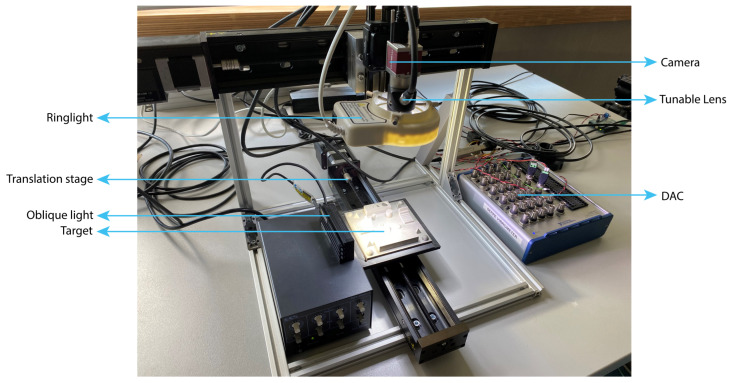
Measurement setup: The object is placed on a translation stage and is illuminated by two lights, the object is recorded by a camera with a tunable lens and microscope objective while moving.

**Figure 4 sensors-22-09805-f004:**
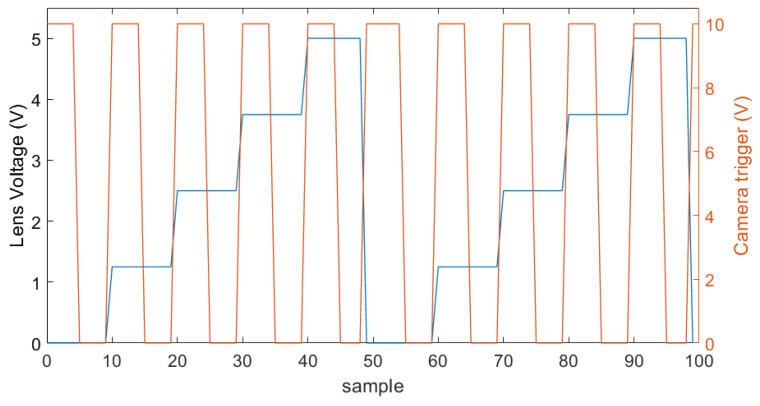
An example of a trigger signal to the camera (right axis, orange) on top of the analog staircase signal to the tunable lens (left axis, blue). The camera is triggered on the falling slope, which in itself is approximately in the center of the voltage step of the tunable lens. This example would capture 5 frames at 5 different focus distances two times.

**Figure 5 sensors-22-09805-f005:**
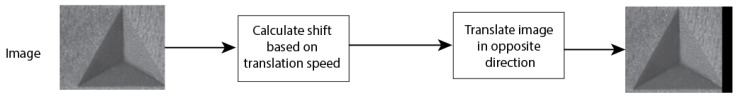
Alignment process of the focal stack, based on translation speed of the camera.

**Figure 6 sensors-22-09805-f006:**
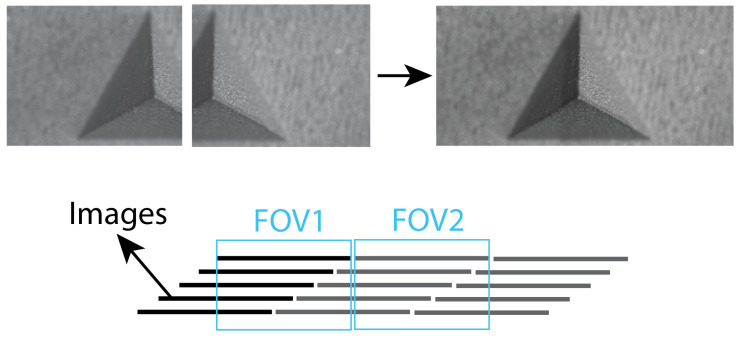
Stiching images of multiple stacks using phase correlation before focus measuring for full FOV measurements.

**Figure 7 sensors-22-09805-f007:**
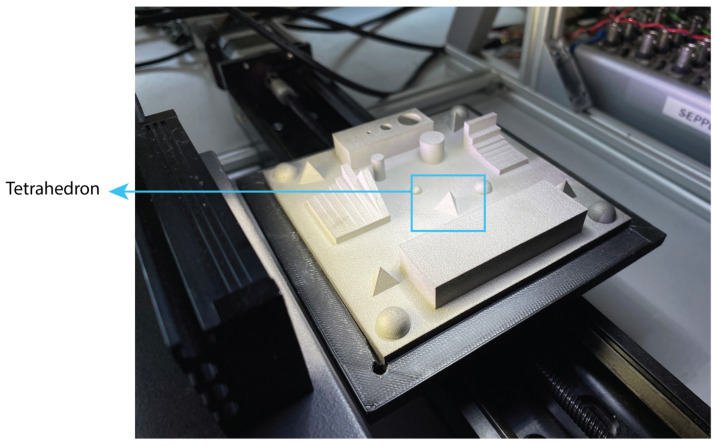
The calibration target and the tetrahedron feature to be measured using both methods.

**Figure 8 sensors-22-09805-f008:**
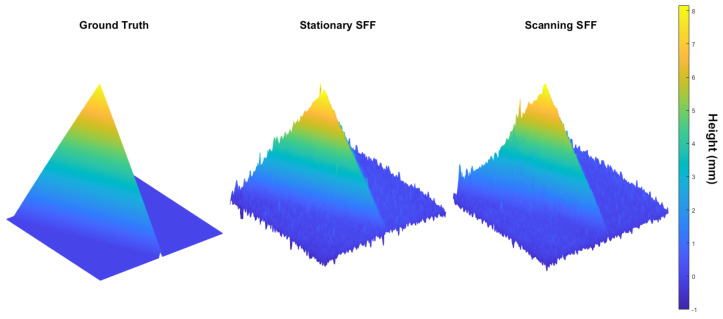
Depth -map of the CAD design of the PCT and the recovered depth-maps using the stationary SFF method and the proposed scanning SFF method.

**Figure 9 sensors-22-09805-f009:**
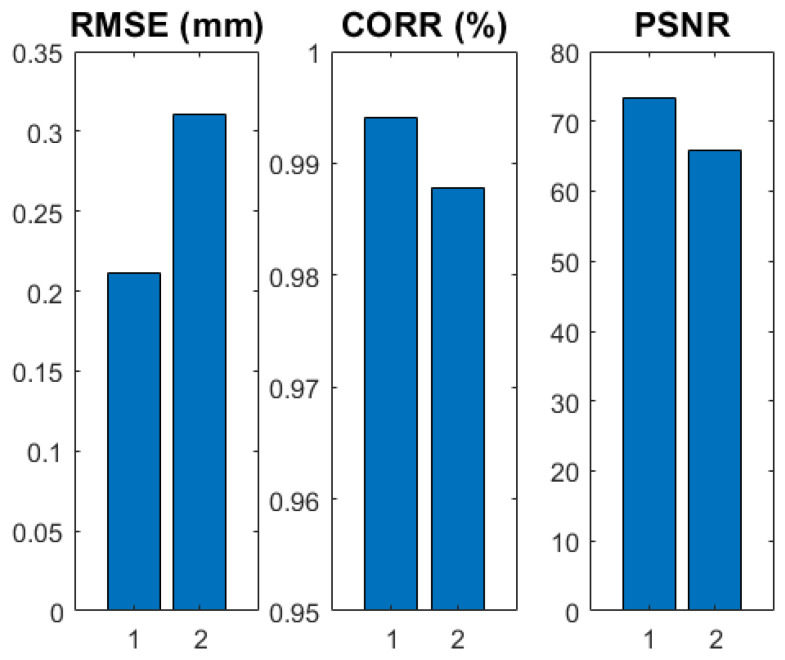
Analytical results of average of 10 comparison measurements between the stationary SFF measurement (1), the scanning SFF measurement (2) and the ground truth.

**Figure 10 sensors-22-09805-f010:**
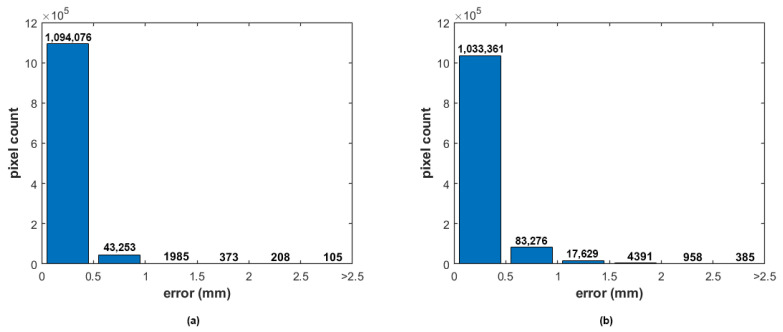
The distribution of errors for each measurement point for both the stationary (**a**) and scanning SFF (**b**) method. It also shows the degradation caused by the measurement method.

**Figure 11 sensors-22-09805-f011:**
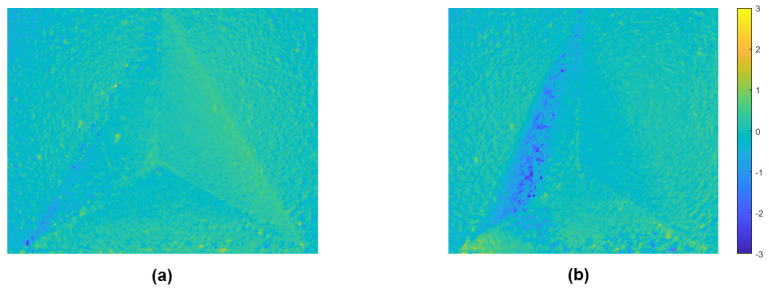
The error maps of both measurements, scaled between −3 mm and 3 mm deviation. (**a**) stationary measurement; (**b**) scanning measurement.

## Data Availability

The data presented in this study are available on request from the corresponding author.

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
