# Peer review of "A Continuous Motion Shape-from-Focus Method for Geometry Measurement during 3D Printing"

_sensors, 2022, doi:10.3390/s22249805_

Round 1
Reviewer 1 Report
The paper sounds promissing and the topic of the paper is interesting in terms of in-line control related to the 3D printing process in principle, but few things must be much better clarified / discussed / underlined in the paper in my opinion, such asȘ
1. The title of the paper in my opinion is not at all very clear / it is incomplete -"A method for continuous motion shape from focus"????? What method? Focus of what? I think that the title must be re-formulated such as to be more specifically related to 3D printing process that is considered in the research that has been analyzed...At some point it is mentioned that this 3D printing method is pwder bed fusion.
2. Even if it is not maybe the primary topic of the paper, in my opinion it is mandatory to create / include a sub-section of the section 2 (my suggestion is this should be the first) in which there must be presented a series of data that are highly important for the research - data about CAD part dimensions (to be measured), aspects related to the 3D printing process itself (type of equipment that was used for printing the part, type of material, 3D printing parameters / orienting of the part, etc.). Does these parameters have a direct influence on the measurements that are done in some way or not? Why yes or why not?
3. In terms of measuring process itself are there any limitations? For instance are there any limitations regarding some materials (e.g. the metallic ones that can create some issues or noise problems in the images)??? Are there any limitations given by the shape of the CAD model (for instance if the part is design in such way that it contains undercuts)? What about the other similar processes (different types of machines that are using the same technology - powder bed fusion) - are there any differences in terms of expected results / are there any limitations caused by the 3D printing method that it is used or for instance if the technolgies are different (powder bed fusion and fused deposition modeling) - can we expect that the proposed method will run the same or not? I haven't seen anywhere such aspects being discussed somehwere in the paper! Maybe it is a room to introduce such sub-section for discussions at the end of Section 3. Personally I would move lot of things that are given in the conclusion section in the discussion sub-section here. Some references to other similar results (even if other researchers have used different method for in-line ontrol than the proposed one) could be introduced / discussed here. This approach will increase the overall scientific level of the paper in the end in my opinion.
4. References must be updated - there is only one reference dated in 2021 and only one in 2022 (out of a total of 26). I personally would include maybe half of the references to be dated in 2021 / 2022 - this will change the impression (that could be of course wrong) that almost nothing was done in the direction of the presented research in 2021 / 2022 - I suppose that this is not true / topic is in the trend with the other current interest / research topics in terms of in-line control / 3D printing methods.
5. Conclusions section must not be a repetition of what has been done / presented in the paper...but must be focused on the achieve results. There are some solutions presented but quite evasive related to what can be improved in the future. Maybe much more details / much concrete and applicative must be provided at this section. Lot of things are related to hardware improvements, but nothing can be done to improve somehow the achieved results from the CAD point of view / software control?
6. There are minor editing errors / spelling and grammar errors in the text - (e.g. in line 29 "surfacel" I think that must be replaced by "surface" / "and and" in line 60 must be replaced by "and"..."To Align" (line 112) must be replaced by "To align"....Personally I would search a synonim also for "abberations" (Line 134 / 147) since I don't like the scientific soundness of this word ("deviation" could be a good synonim) - this is just a suggestion
Overall I appreciate that this paper has good potential and brings an interesting new approach for the in-line control of the 3D printing processes of different parts even if the above details (important in my opinion) about 3D printing process that was used / limitations of the proposed method / comparison with other researchers' results, updating of some references are needed to be brought by the authors in the next period, before the paper could be considered for being published in the Sensors MDPI journal.
Since data about 3D printing process is missing completely in the paper (in this variant at least) this is the main reason why I have selected that the paper must be reconsidered after major revision"...since this aspect is really crucial in my opinion. In-line measurement done must be correlated by the 3D printing process itslelf, type of material and type of the CAD part that is being produced / measured / evaluated by using this method in the end.
Reviewer 2 Report
This paper presents a novel methodology by using shape from focus in a continuous scanning motion, thus supplies a measurement for 3D print manufacturing et al. The experiments validate the idea. However, the systematic design process regarding the particular need should be elaborated in more details.
Reviewer 3 Report
The manuscript describes an application of the Shape from focus methods applied to movable object shape measurement. The content can be interesting for readers, but quality of presentation has to be improved.
Besides some type and context errors I miss any mathematical description of the problem. There are evaluated some parameters within the text, but these sentences need to be more clearly explained with mathematical relations showing how the resolution is linked with frame rate and number of images and other parameters! All calculation expressed in the text of section 2 should be based on expressed equations.
Equations should also express speed and camera parameters necessary for „translating over slightly less than 50% of the FOV, there will be some overlap between images of consecutive stacks.“
There is also not satisfying explained methodology of data error evaluation. Are you able to separate error caused by the 3D printing and error caused by the optical measurement method? You decided to use statistical methods, which minimize effect of extreme deviations. These extreme deviations are much more important for practical consideration of the 3D printing fidelity than „average“ deviations. Have you analyzed influence of the measurement methods at different parts of the sample? It seems the RMS of the bottom surface is much greater than the inclined surfaces.
In my point of view the manuscript without equations explaining relation among individual variables of the problem don´t help reader to repeat described process. Similar situation is in part describing the experiment. Why reducing the resolution of the images to a quarter is good enough? Why other values or image reduction are not right – what is the optimum image resolution reduction and how it can be found?
Discussion does not contain what are critical details of the 3D printed parts nor where the measurement method provides the largest deviations?
Round 2
Reviewer 1 Report
The details that were introduced in the new variant of the revised paper and the explanations provided in the cover letter were helpful for a better understanding of the process and the limitations reqarding the presented method as well. In the new variant of the paper the reformulated it is much better. Overall I personally consider that the authors did all their best to do the required corrections and therefore I agree with publishing of the paper in the current form with some minor changes that I kindly recommend to the authors to change (if it is possible):
- personal way of addressing maybe could be avoided in the text - instead of "we can see" maybe the expression could be replaced with "as one may notice"...and in the Conclusions section where it is stated "We have introduced a new method" maybe it is better to replace this with "One new method was proposed and introduced...."....This is just a suggestion / recommendation (which is minor) but could be taken into consideration if the authors are considering it (as my previous recommendations and requests) being constructive.
Overall I have appreciated all the corrections / improvements that were made as suggested (including even the References which were also updated)!
Congratulations to the authors for the current work and perspectives of their research that they are seeing it for the future!
Author Response
Thank you for your kind words,
The usage of the word "We" was reduced in the whole manuscript.
Lines: 44, 50-51, 56-57, 79, 91, 110, 112, 115-116, 138, 139-140, 179, 188, 190-191, 208 and 218 updated
Reviewer 3 Report
The revised manuscript is well upgraded and clarified. There are still some type errors as in lines 30 and 109. There is not explained meaning of abbreviation GT used in line 198. But overall I agree with paper publication.
Author Response
Thank you for your kind words,
The suggested type errors were updated and in addition the whole document was analysed for grammar mistakes and type errors. All mistakes were corrected.
The missing abbreviations were added to the abbreviation list including ground truth (GT)